# Tocotrienol-Rich Fraction and Levodopa Regulate Proteins Involved in Parkinson’s Disease-Associated Pathways in Differentiated Neuroblastoma Cells: Insights from Quantitative Proteomic Analysis

**DOI:** 10.3390/nu14214632

**Published:** 2022-11-03

**Authors:** Kasthuri Bai Magalingam, Premdass Ramdas, Sushela Devi Somanath, Kanga Rani Selvaduray, Saatheeyavaane Bhuvanendran, Ammu Kutty Radhakrishnan

**Affiliations:** 1Jeffrey Cheah School of Medicine and Health Sciences, Monash University, Bandar Sunway 47500, Malaysia; 2School of Postgraduate Studies, International Medical University, Kuala Lumpur 57000, Malaysia; 3Division of Applied Biomedical Sciences and Biotechnology, School of Health Sciences, International Medical University, Kuala Lumpur 57000, Malaysia; 4Pathology Division, School of Medicine, International Medical University, Kuala Lumpur 57000, Malaysia; 5Product Development and Advisory Services Division, Malaysian Palm Oil Board, Bandar Baru Bangi 43000, Malaysia; 6Monash-Industry Palm Oil Education and Research Platform (MIPO), Monash University Malaysia, Bandar Sunway 47500, Malaysia

**Keywords:** tocotrienol-rich fraction (TRF), SH-SY5Y neuroblastoma cells, Parkinson’s disease, proteomics

## Abstract

Tocotrienol-rich fraction (TRF), a palm oil-derived vitamin E fraction, is reported to possess potent neuroprotective effects. However, the modulation of proteomes in differentiated human neuroblastoma SH-SY5Y cells (diff-neural cells) by TRF has not yet been reported. This study aims to investigate the proteomic changes implicated by TRF in human neural cells using a label-free liquid-chromatography-double mass spectrometry (LC-MS/MS) approach. Levodopa, a drug used in the treatment of Parkinson’s disease (PD), was used as a drug control. The human SH-SY5Y neuroblastoma cells were differentiated for six days and treated with TRF or levodopa for 24 h prior to quantitative proteomic analysis. A total of 81 and 57 proteins were differentially expressed in diff-neural cells following treatment with TRF or levodopa, respectively. Among these proteins, 32 similar proteins were detected in both TRF and levodopa-treated neural cells, with 30 of these proteins showing similar expression pattern. The pathway enrichment analysis revealed that most of the proteins regulated by TRF and levodopa are key players in the ubiquitin-proteasome, calcium signalling, protein processing in the endoplasmic reticulum, mitochondrial pathway and axonal transport system. In conclusion, TRF is an essential functional food that affects differential protein expression in human neuronal cells at the cellular and molecular levels.

## 1. Introduction

TRF, a palm oil extract, consists of 25% α-tocopherol (α-Toc) and 75% tocotrienols (T3). The tocotrienols that exist in TRF are a mixture of four naturally occurring isoforms i.e., alpha-T3 (α-T3), beta-T3 (β-T3), delta-T3 (δ-T3) and gamma-T3 (γ-T3). The T3 isoforms contain a chromanol ring with a hydroxyl (-OH) group that can donate a hydrogen (H) atom to counteract the deleterious effects of free radicals and a hydrophobic side-chain that can readily permeate cell membranes. In addition, the presence of three double bonds at positions 3′, 7′ and 11′ in the side-chain of the T3 may contribute to the superior antioxidant potency compared to the Toc isoforms [1]. Vitamin E accumulates in lipid-rich regions of cells such as organelles, cell membranes, mitochondrial membranes and lipoproteins. Studies have found α-Toc to be a potent antioxidant molecule that can break the propagation of the peroxyl chain in redox cycle reaction [2,3]. T3 and α-Toc in TRF may amplify the beneficial changes at cellular and molecular levels and result in disease prevention in humans.

Several studies have investigated the neuroprotective effects of TRF using toxin-induced damage in cell-based and animal models. TRF attenuated glutamate-induced injury in astrocytes by reducing mitochondrial damage and increasing the number of viable cells [4]. In a mouse model of Alzheimer disease (AD), TRF supplementation modulated the hippocampal gene expression affecting the biological process and pathways that delay the pathological changes [5]. In addition, TRF also regulated the metabolic pathways that improved exploratory activity, spatial learning and memory in mice [6]. In rats, TRF supplementation caused significant changes in L-arginine metabolism and age-associated biochemical changes in the entorhinal cortex and cerebellum, indicating the enhancement of memory and motor functions [7]. A recent clinical study also found that supplementation with TRF for six months or more significantly increased the activity of antioxidant enzymes, such as superoxide dismutase (SOD) and glutathione peroxidase, and the ratio of reduced glutathione (GSH)-to-oxidized glutathione (GSSG) in healthy individuals between 50–55 years old [8]. In another clinical study, stroke patients supplemented with TRF showed a remarkable reduction in the mean white matter lesion (WML) size at the end of two years, compared to placebo groups [9]. Although these studies have delineated the protective effects of TRF using various biochemical and clinical approaches, studies on the proteomic changes implicated by TRF at the molecular and cellular level in PD remain scarce.

A recent quantitative proteomic study reported that TRF supplemented AβPP/PS1 mice altered the expression of proteins, which improved cognitive functions and modulated amyloid pathology in the hippocampus, medial prefrontal cortex and striatum of these mice [10]. Given the lack of cell-based proteomics data on the effects of TRF in PD, we embarked on a comparative proteomic analysis to discover differential proteins altered by TRF in comparison with levodopa. We also investigated the common and exclusive pathways modulated by TRF and levodopa in diff-neural cells. Furthermore, we performed the comparative functional bioinformatics analyses on differentially expressed proteins between TRF and levodopa treatments on diff-neural cells. We evaluated the proteins targeted by TRF and/or levodopa that play vital roles in various cellular processes and signalling pathways that protect the neural cells. As levodopa is often prescribed to PD patients as a dopamine replacement agent, we have intricately scrutinized and compared the alteration in protein expression implicated by TRF and levodopa on PD pathway in neuronal cells using the online bioinformatic Kyoto Encyclopaedia of Genes and Genomes (KEGG) pathway database resource [11].

## 2. Materials and Methods

### 2.1. Reagents

The SH-SY5Y neuroblastoma cells (Cat# CRL-2266) were purchased from the American Tissue Culture Collection (ATCC), and cultured in Dulbecco Modified Eagle Medium (DMEM) (Biosera, Nuaille, France) supplemented with 4.5 g/L glucose (Biosera, Nuaille, France) and L-glutamine without sodium pyruvate (Biosera, Nuaille, France) and 10% fetal bovine serum (FBS) (Biosera, Nuaille, France). Penicillin-Streptomycin solution (P/S) and non-essential amino acid (NEAA) were obtained from Gibco, Carlsbad, CA, USA. Retinoic acid (RA), dimethylsulphoxide (DMSO), 3,4-Dihydroxy-L-phenylalanine (levodopa) were purchased from Sigma Aldrich, St. Louis, MO, USA. The tocotrienol-rich fraction (TRF) was a kind gift from Excelvite Sdn. Bhd, Ipoh, Malaysia. The EasyPrep Mini MS Sample Prep kit and Pierce BCA Protein Kit were obtained from Thermo Fisher Scientific, USA. For the PCR validation assay, the commercial primers i.e., All-in-One qPCR Primers for the human CALM1 (NM_006888.5), TUBB (NM_178014.3), ATP5F1A (NM_004046.5), UCHL1 (NM_0044181.4) and GAPDH (NM_002046.6) genes were purchased from Genecopoeia, MD, USA. The HiScript II One Step qRT-PCR SYBR Green Kit was purchased from Vazyme biotech Co., Ltd., Nanjing, China.

### 2.2. Cell Culture and Differentiation of SH-SY5Y Human Neuroblastoma Cells

The SH-SY5Y human neuroblastoma cells were cultured in culture medium consisting of DMEM, 10% heat-inactivated FBS (HI-FBS), 1% P/S and 1% NEAA at 37 °C in a humidified 5% CO_2_ incubator. Culture media was replaced every alternate day, and cells were sub-cultured when it reached 70% confluence. To differentiate these human neuroblastoma cells, 1 × 10^6^ SH-SY5Y cells were seeded in complete culture medium in a T75 flask. After 24 h, the culture medium was replaced with fresh medium that consist of DMEM supplemented with 3% HI-FBS, 1% P/S, 1% NEAA and 10 μM retinoic acid. The cells were allowed to differentiate for six-days with the medium replaced once on day 4. After six days, the terminally differentiated SH-SY5Y neural cells (diff-neural cells) expresses the dopaminergic features at morphological, biochemical and genetic levels [12].

### 2.3. Treatment Protocol

On day 7, the culture medium was removed and replaced with serum-free medium containing TRF (0.1 μg/mL) or levodopa (0.1 μg/mL) and the diff-neural cells were incubated at 37 °C for 24 h in a humidified 5% CO_2_ incubator. The concentration (0.1 μg/mL) of TRF and levodopa were determined from preliminary cytotoxicity studies using a range of concentrations (0 to 20 μg/mL). After 24 h of exposure to either TRF or levodopa, the diff-neural cells were harvested and subjected to protein extraction. The untreated cells were simultaneously maintained in a serum-free culture medium and served as controls.

### 2.4. Protein Extraction, Reduction and Alkylation

Protein extraction and processing to yield MS-ready peptides was achieved using the EasyPep Mini MS Sample Prep kit. The treated (levodopa or TRF) and untreated diff-neural cells were centrifuged for 7 min at 120 g, and supernatants were discarded. Then, 100 μL of lysis buffer (*provided with the kit*) and 1 μL of Universal Nuclease (*provided with the kit*) were added to the pelleted cells and mixed vigorously until sample viscosity was reduced. The protein concentration was determined using Pierce BCA Protein Assay Kit. Then, lysis solution (*provided with the kit*) was used to prepare the final protein concentration of 100 μg for the protein reduction step.

A volume of 50 μL of Reduction Solution (*provided with the kit*) was added to the sample and gently mixed. This was followed by 50 μL of Alkylation solution (*provided with the kit*), again with gentle mixing. Following this, the samples were incubated at 95 °C using a heat block for 10 min to reduce and alkylate the samples. At the end of this incubation, the samples were cooled to room temperature.

### 2.5. Protein Digestion and Peptide Clean-up

In the protein digestion step, 50 μL of Trypsin/Lys-C Protease enzyme (provided with the kit) was added to the reduced and alkylated protein samples and incubated at 37 °C for 3 h with intermittent shaking. Then, 50 μL of digestion stop solution (provided with the kit) was added to the sample and mixed gently. The digested protein samples were transferred to individual Peptide Clean-up columns (provided with the kit) and centrifuged at 1500× *g* for 2 min. Then, 300 μL of Wash Solution A (provided with the kit) and Wash Solution B (provided with the kit) were sequentially added to the column and centrifuged (1500× *g* for 2 min). Next, the Peptide clean-up columns were transferred into clean 2 mL microcentrifuge tubes, and 300 μL of Elution buffer (provided with the kit) was added to these columns, and the tubes were centrifuged. Finally, the peptide samples were dried using a vacuum centrifuge for 4-h and stored at −80 °C prior to the LCMS/MS analysis. Prior to the LCMS/MS analysis, the dried samples were resuspended in 100 μL of 0.1% formic acid in water.

### 2.6. Liquid Chromatography Electrospray-Ionization Coupled with Tandem Mass Spectrometry Analysis

Proteomic analysis was conducted using LC-MS/MS according to the methods described previously [13]. Upon injection of 1 μL of 100 μg of extracted peptides suspended in 0.1% formic acid into the LC-MS/MS analyser, the samples underwent chromatographic separation in Agilent Large Capacity Chip, 300 Å, C18 (Agilent, Santa Clara, CA, USA) through a 75 µm × 150 mm analytical column. The samples were separated using a gradient mobile phase system with a total running time of 47 min and a constant flow rate of 4 μL/min from Agilent 1200 Series Capillary pump and 0.5 μL/min from Agilent 1200 Series Nano pump. The autosampler temperature was set at 4 °C. The quadrupole-time of flight (Q-TOF) polarity was set at positive with capillary and fragmenter voltage at 1900 V and 360 V, respectively, and 5 L/min of gas flow with a temperature of 325 °C. The collision energy was determined at 3.7 V (100 Da), and reference masses with positive polarity was set at 299.294457 and 1221.990637. The peptide spectrum was analysed in auto MS mode ranging from 110–3000 m/z for MS scan and 50–3000 m/z for MS/MS scan.

### 2.7. Data Computation and Protein Identification

The mass spectra data were analysed with Agilent MassHunter (Agilent, Santa Clara, CA, USA) data acquisition software and PEAKS X 7.0 software [14] was used to identify proteins against Homo sapiens sequences obtained from Uniprot, Swissprot and TrEMBL databases. Trypsin was selected as the digestive enzyme, while the maximum missed cleavage by the enzymatic digestion was set at three sites for the protein sequence with carbamidomethylation set as a fixed modification. The protein identification was filtered using a target-decoy approach with a global false discovery rate (FDR) of 1% determined by the in-built FDR tool within PEAKSX+ software. The unique peptide ≥1 was used to filter out inaccurate proteins. The filtration parameter of protein abundance score of −10lgP ≥ 20 and peptide −10lgP ≥15: the identified proteins are relatively high in confidence as it targets very few decoy-matches above that threshold.

### 2.8. Differential Protein Expression Analysis

The raw mass spectra data obtained from MS/MS analysis were visualized using PEAKS X+ and the differentially expressed protein between two treatment groups were evaluated and corroborated with Uniprot search engine by species selection, i.e., Homo sapiens. The protein data were filtered for three valid values from three biological replicates, and proteins that were only presented in one or two biological replicates were eliminated. The biological replicates from all samples were clustered under the same matrix, and the outliers were removed, and the missing data was assigned with a random number derived from a normal distribution. The TRF-treated protein sets were compared to the protein sets of untreated controls using a two-tailed, Student *t*-test to identify significant proteins altered by TRF in diff-neural cells. The proteins were regarded as significantly regulated if the difference between the two groups showed a *p*-value of less than 0.05 (*p* < 0.05). Similar computation was performed on levodopa-treated cells against untreated control cells. The significantly upregulated and down-regulated proteins in response to TRF or levodopa treatment with average mass, *p*-value and log2 fold expression are listed in Table 1 (TRF) or Table 2 (Levodopa).

### 2.9. Determination of Common and Unique Proteins between TRF and Levodopa Treatment

A Venn diagram was generated to identify unique and overlapping proteins between the differentially expressed proteins (*p* < 0.05) in TRF-treated vs. untreated control or levodopa-treated vs. untreated control using an online software (http://bioinformatics.psb.ugent.be/webtools/Venn/ (accessed on 5 April 2021)). The unique proteins regulated by TRF or levodopa represent distinct pathways regulated by these compounds in diff-neural cells. The overlapping proteins indicate the mutual biological pathways mediated by TRF and levodopa in neuronal cells.

### 2.10. Gene Ontology (GO) Term and Pathway Enrichment Analysis

The functional GO annotations of the significantly expressed proteins were analysed using the PANTHER version 16 (http://pantherdb.org (accessed on 24 September 2021)), and the proteins were classified based on their molecular function, biological process and cellular components. The protein classification was analysed by uploading the differentially expressed proteins regulated by TRF or levodopa in the PANTHER GO term interface. Only the top 10 GO terms that showed the highest fold enrichments for each classification was analysed. All results shown expressed adjusted *p*-value < 0.05 as determined by the Fischer’s Exact test and False Discovery Rate. We also identified the functional protein class of the 32 overlapping proteins between TRF and levodopa using the PANTHER database. The Pathway enrichment analysis was performed using the KEGG (Kyoto Encyclopaedia of Genes and Genomes) https://www.genome.jp/kegg/pathway.html (accessed on 20 October 2021) database for the differentially expressed proteins regulated by TRF or levodopa. We highlighted the top 5 enriched pathways implicated by TRF or levodopa treatment on diff-neural cells. The differentially expressed proteins regulated by TRF or levodopa enriching the KEGG-Parkinson’s disease pathway. Besides applying the bioinformatics platform to understand the biological roles of the expressed proteins, cross-referencing with published reports was also conducted to validate these proteins’ functions.

### 2.11. Validation of mRNA by One Step Quantitative PCR (qPCR) Assay

Four candidate genes were chosen from each PD pathway as curated by KEGG to validate the proteomic data using qPCR technology. The four genes are responsible for coding proteins that play a pivotal role in the PD pathway, i.e., CALM1 protein in the calcium signalling pathway, ATP5F1A in the mitochondrial pathway, TUBB in the axonal transport system and UCHL1 in the ubiquitin pathway. Purified RNA was processed using the HiScript II One Step qPCR SYBR Green Kit according to the manufacturer’s instruction. The total reaction volume was 20 μL consisting of 100 pg of RNA, 10 μL of 2x One Step SYBR Green Mix, 1 μL of One Step SYBR Green Enzyme Mix, 0.4 μL of 50x ROX Reference Dye and 2 μL of forward and reverse primers (10 μM) of CALM1, TUBB, ATP5F1A, UCHL1 or GAPDH genes. The assay was conducted in One Step qRT-PCR protocol commencing with reverse transcription at 50 °C for 3 min, followed by pre-denaturation at 95 °C for 30 s. After this step, the samples went through 40 cycles involving denaturation at 95 °C for 10 s, followed by 30 s at 60 °C of extension. A dissociation curve analysis was performed at the end of the cycle to confirm amplification specificity. All samples were run in triplicates. The signal was detected using ABI StepOne Plus (ThermoFisher Scientific, Waltham, MA, USA). The CALM1, TUBB, ATP5F1A and UCHL1 messenger RNA (mRNA) levels were normalised to the mRNA of the reference gene GAPDH. The non-template control (sample without RNA template) was run parallel in triplicate for each primer to assess for non-specific amplification.

### 2.12. Statistical Analysis

Statistical analysis comparing the quantitative proteomic data from TRF-treated vs. untreated control cells and levodopa-treated vs. untreated controls in diff-neural cells were performed using a two-tailed Student’s *t*-test. All statistical analyses were performed with SPSS v20. Differentially expressed proteins that displayed the difference in Log2 Fold change with *p* < 0.05 were regarded as statistically significant. For the qPCR validation assay, the treatment groups (TRF, levodopa) were compared with the untreated group using the Bonferroni multiple-comparison post-hoc test. A value of *p* < 0.05 was considered statistically significant.

## 3. Results

### 3.1. The Effects of TRF and Levodopa Treatments on Total Protein Extract from Diff-Neural Cells

We conducted a shotgun proteomic investigation using label-free tandem liquid mass spectrometry (LC-MS/MS) on TRF or levodopa’s treated diff-neural cells. The pre-processing of raw tandem mass spectrometry data was conducted using PEAKS X+ to explore the alterations in diff-neural cells’ proteome profile in response to TRF or levodopa treatment. A total of 528 statistically significant proteins were identified from a total of 3828 proteins in three biological replicates of untreated control cells. As from TRF treatment on diff-neural cells, a sum of 4019 proteins were detected in all three biological replicates that constituted 707 common protein data sets. A total of 163 proteins from TRF treated diff-neural cells matched with protein sets from untreated control cells. Among these 163 proteins, 81 proteins showed a significant differential regulation, with 53 upregulated and 28 down-regulated (Table 1).

Meanwhile, the levodopa treated diff-neural cells yielded a total of 3603 proteins, with 682 proteins distinct all three biological replicates. When we compared the common protein sets between levodopa treated and untreated control cells, we found a total of 166 proteins that matched both the data sets with 57 proteins expressing a significant difference in expression with *p* < 0.05. Among the 57 significantly expressed proteins, 32 proteins demonstrated a significant upregulation, while 25 were down-regulated (Table 2).

### 3.2. Venn Diagram Analysis

The differentially expressed proteins regulated by TRF and levodopa treatments were further stratified using a Venn diagram to identify the unique and overlapping proteins. The overlapping regions illustrate the proteins’ involvement in common biological and molecular processes in TRF and levodopa treated cells. The Venn diagram revealed 49 and 25 unique proteins regulated by TRF and levodopa, respectively. Importantly, a total of 32 differentially expressed proteins were common to both TRF and levodopa treatments (Figure 1). The list of unique and overlapping proteins expressed in response to TRF or levodopa treatment in diff-neural cells are shown in Table 3.

### 3.3. Functional Gene Ontology (GO) Enrichment Analysis

The GO enrichment analysis identifies specific genes or proteins’ biological and functional roles. To further understand the alterations in cellular and molecular processes regulated by TRF or levodopa treatments in diff-neural cells, the differentially expressed proteins (*p* < 0.05) were mapped to their enriched functional annotation GO terms from the PANTHER bioinformatics database. The functional annotations of differentially expressed proteins modulated by TRF or levodopa compared against untreated controls were classified into three classes-molecular function, biological processes and cellular components, as shown in Figure 2A,B.

A total of nine GO terms biological processes were significantly overexpressed (>100) in TRF implicated diff-neural cells. Examples of these GO term biological processes are endodeoxyribonuclease activity, DNA topoisomerase (ATP hydrolysing) activity and CRD-mediated mRNA stabilization. As for the molecular function, TRF implicated differentially expressed proteins modulated the GO term classes of structural constituents of the postsynaptic actin cytoskeleton, peroxiredoxin activity and type 3 metabotropic glutamate receptor biding. These molecular and biological processes were found to be highly prevalent at cellular component regions, particularly PTW/PP1 phosphatase complex, eukaryotic translation elongation factor 1 complex and CRD-mediated mRNA stability complex, as displayed in Figure 2A.

A total of three GO terms biological processes exhibited a significant fold change overexpression (>100) in levodopa implicated diff-neural cells. Examples of these GO term biological processes are protein folding in endoplasmic reticulum, retinoic acid biosynthetic process and transepithelial process. As for the molecular function, levodopa implicated differentially expressed proteins modulated the GO term classes of structural constituents of the postsynaptic actin cytoskeleton, L-lactate dehydrogenase activity and lactate dehydrogenase activity. These molecular and biological processes were found to be highly prevalent at cellular component regions, particularly endoplasmic reticulum chaperone complex and unfolded protein binding, as displayed in Figure 2B.

### 3.4. Common Protein Expression between TRF and Levodopa Treatment in Diff-Neural Cells

The treatment of TRF or levodopa on diff-neural cells yielded 32 significant common proteins (*p* < 0.05). The fold expression patterns of these 32 common proteins between TRF vs. untreated and levodopa vs. untreated diff-neural cells are illustrated in Figure 3A. Among the 32 proteins, 30 proteins regulated by TRF displayed similar expression pattern with the levodopa in diff-neural cells. In contrary, two out of the 32 proteins displayed an opposite expression pattern between levodopa and TRF: MYL6 and VGF. The similar protein expression patterns regulated by TRF and levodopa treatments indicate the enrichment of common neuroprotective pathways in diff-neural cells.

Using the PANTHER online database, 28 out of the 32 proteins grouped into 10 functional categories; cytoskeletal protein (TUBB, MYL6, TUBB2B, ACTB, TUBB4B, ACTG1, ACTC1, TUBB6, TUBA1B, TUBB3), chaperone (PPIA, P4HB, NPM1, NASP, CANX, HSP90AB1), chromatin/ chromatin-binding or regulatory protein (HIST1H1B, H2AFX), intercellular signal molecule (VGF), metabolite interconversion enzyme (PRDX2, LDHA), nucleic acid metabolism protein (FUS), protein modifying enzyme (UCHL1), protein-binding activity modulator (PPP2R1A), translational protein (RPS3, RPS16, RPLP1) and transporter (ATP5F1A) as shown in Figure 3B. The four proteins that did not belong to any protein classes are CRABP2, HNRNPU, H2AFX, HIST1H1C).

### 3.5. Pathway Enrichment Analysis of the Differentially Expressed Proteins

The KEGG database provides mechanistic insights into biological pathways enriched by differentially expressed protein set following treatment conditions. The pathway construction is based on the molecular interaction network of how genes/proteins interact and regulate. When 81 and 57 differentially expressed proteins implicated by TRF and levodopa, respectively, were uploaded in the KEGG mapper, we identified the top five enriched neurodegenerative disease pathways as shown in Figure 4. These five significantly enriched pathways modulated by TRF or levodopa in diff-neural cells are pathways associated with neurodegenerative disease-multiple disease (hsa05022), PD (hsa05012), Amyotrophic lateral sclerosis (hsa05014), AD (hsa05010) and Huntington’s disease (hsa05016). The identity of the differentially expressed proteins enriching the respective pathways modulated by TRF or levodopa are listed in Table 4.

### 3.6. TRF and Levodopa Implicated Differential Protein Expression in PD Pathways

The KEGG-PD pathway depicts the consolidation of multiple pathways that lead to dopaminergic neuronal demise. The four vital pathways that cause neurodegeneration are ubiquitin, calcium, mitochondrial, and axonal transport defects (red box). Diff-neural cells treated with TRF and levodopa resulted in upregulation of UCHL1 protein in the ubiquitin pathway. Moreover, TRF upregulated the CALM1, CALM2 and CALM3 [CaM] proteins in the calcium signalling pathway and downregulated the HSPA5 [Bip] in protein processing the endoplasmic reticulum. TRF and levodopa upregulated the ATP5F1A [CxV] in the mitochondrial pathway. While SLC25A6 [mPTP] was upregulated by TRF and VDAC1 and VDAC3 [mPTP] were significantly suppressed by levodopa. Notably, both TRF and levodopa upregulated the TUBA1A, TUBA1B, TUBB, TUBB2B, TUBB3, TUBB4B, TUBB6 [TUBA, TUBB] that mediate the Axonal transport system. The effects of TRF (T) and levodopa (L) in regulating differential protein expression in KEGG-PD pathways are illustrated in Figure 5.

### 3.7. Validation of Protein Expression Using qPCR Assay

The mRNA expressions of four selected genes were investigated using the One Step qPCR assay to validate the proteomic data obtained using the LC-MS/MS assay. As shown in Figure 6, the qPCR data correlated with proteomic findings with all four genes displayed similar expression patterns as respective protein expression. The ATP5F1A, UCHL1 and TUBB proteins were significantly upregulated in TRF and levodopa treated diff-neural cells compared to untreated control cells. Although CALM1 protein expression was upregulated in both TRF and levodopa treated neural cells, but the significant difference with at least *p* < 0.05 was only demonstrated by TRF (Figure 6A).

In the qPCR assay, TRF upregulated the mRNA expression of ATP5F1A, UCHL1 and TUBB with *p* < 0.05. However, the CALM1 upregulation by TRF was not statistically significant. Levodopa showed a remarkable overexpression of ATP5F1A, CALM1, UCHL1 and TUBB with *p* < 0.05 (Figure 6B).

## 4. Discussion

The present study was aimed to provide insights into TRF modulated proteome footprints in diff-neural cells using an in vitro model of human dopaminergic neural cells. We compared these findings with proteomes regulated by levodopa, a standard drug used in the treatment of PD. The diff-neural cells were established by culturing cells in retinoic acid and low-serum culture medium for 6 days. The established diff-neural cells exhibit neuronal characteristics such as suppression of proliferation rate, generation of long and branched neurites and secretion of neuronal marker proteins [15]. Our previous study has shown that diff-neural cells express characteristics of dopaminergic cells at morphology, biochemical and genetic levels. This study demonstrated that differentiated human neural cells are the most appropriate platform for understanding the effects of potential drugs targeting the various cellular and molecular signalling processes [12]. In the present quantitative proteomic study, we discovered that TRF and levodopa treatment on diff-neural cells significantly regulated 81 and 57 proteins, respectively. TRF treated diff-neural cells upregulated 53 proteins and suppressed 28 proteins (Table 1). While levodopa treated diff-neural cells upregulated 32 proteins and down-regulated 25 proteins (Table 2).

In addition, TRF and levodopa treated diff-neural cells significantly upregulated the chaperone proteins, including HSP90AB1, HSPD1 and PPIA (Figure 3A,B). The heat shock proteins (HSP) HSP90AB1(HS90B) and HSPD1(Hsp60) are evolutionarily well-conserved molecular chaperones that assist in protein folding and maintain protein stability. As the accumulation of misfolded proteins can cause deleterious effects to cells, the HSP90AB1 interacts with more than 20 co-chaperones that facilitate recognising unfolded client proteins and uses ATP energy to transform these proteins to their biologically active conformations via hydrolysis [16]. Whereas HSPD1, a nuclear-encoded mitochondrial chaperonin, is readily expressed in the event of oxidative stress, heat shock and DNA damage. In mitochondria, HSPD1 coupled with Hsp10 forms a folding apparatus for the correct folding of mitochondrial proteins [17]. Previous studies have documented that levodopa administration in rats significantly upregulated HSPD1in the substantia nigra region. The over-expression of HSPD1 was in line with dose-dependent increased activity of mitochondrial complex I as well as nitrosative stress in rats [18]. On the other hand, we also showed that TRF and levodopa stimulated the expression of peptidylprolyl isomerase A (PPIA) in diff-neural cells. The PPIA exists in the highest amount in the brain cells and functions as a molecular chaperone and plays a crucial role in the maturation, folding and assembly of neuron-specific membrane proteins. A report from Pasetto et al. [19] has further validated that the intracellular expression of PPIA is beneficial as this enzyme functions as foldase to correct proteins to achieve its normal conformation.

The dopaminergic neurodegeneration at substantia nigra pars compacta (SNpc) and dopamine depletion at striatum have been documented to be the consequences of the impairment of multiple neurophysiological pathways. Genetic mutations, oxidative stress, mitochondrial defects, failure in the clearance of toxic proteins, aggregation of misfolded proteins (Lewy bodies) are some of the reported mechanisms involved in the pathogenesis of PD [20]. In our investigation, we explored the role of TRF in modulating the crucial pathways that may impede the neurodegenerative process in diff-neural cells. On that note, we applied the online bioinformatics database—KEGG, to understand the effects of differential protein expression implicated by TRF on various pathways leading to PD. Here, we report that TRF treatment on diff-neural cells resulted in differential expression of proteins in the ubiquitin pathway, calcium signalling pathway, mitochondrial pathway and axonal transport system (Figure 5).

The ubiquitin pathway is the fundamental enzymatic cascade of reaction that functions in the clearance of impaired and misfolded proteins in nucleus and cytoplasm of eukaryotic cells. Mutation in any component enzymes or proteins of the ubiquitin pathway is linked to the blockade in protein homeostasis leading to proteinopathy and neurodegeneration [21]. The ubiquitin carboxyterminal hydrolases 1 (UCHL1), also known as PARK 5, has been suggested to be involved in monoubiquitin processing via the short C terminal peptide cleavage and ubiquitin recovery from proteasomal degradation protein remnants. These processes maintain the optimal ubiquitin pool to remove damaged proteins in the ubiquitin proteosome pathway (UPP), as depletion in ubiquitin pools has been noted in UCHL1 knockout neurons [22]. A recent study has proposed the different roles of UCHL1 in normal and diseased conditions. According to this study, the suppression of UCHL1 in primary neurons and hippocampal tissues from mice resulting in overexpression and accumulation of α-synuclein in these neurons. On the contrary, inhibition of UCHL1 activity in α-synuclein overexpressing neurons strikingly reduced the α-synuclein level and promoted its clearance at neuron synapses [23]. Our finding shows the UCHL1 protein expression was significantly upregulated following treatment with TRF and levodopa compared to untreated controls. We postulate that the increase in UCHL1 protein expression indicates TRF and levodopa’s neuroprotective actions of in boosting up the monomeric ubiquitin levels and maintaining the normal synaptic structure of the neurons. Our claim is in line with a previous study that showed the correlation between the increased expression of UCHL1 in neurons and restoration of the normal synaptic remodelling through upregulation of free monomeric ubiquitin levels [24]. Therefore, TRF is a safe and effective disease-modifying therapy to be considered in treating neurodegenerative diseases.

Calcium (Ca^2+^) via its signalling pathway in neurons, plays a central role in regulating a plethora of cellular activities, including synaptic plasticity and transmission, differentiation of neurons, gene expression and regulation of complex protein networks [25]. Neurons effectively maintain a steep Ca^2+^ gradient between intra- and extracellular compartments, notably low intracellular Ca^2+^ concentration compared to extracellular space. The alteration in intracellular Ca^2+^ concentration is triggered by the opening of Ca^2+^ channels such as voltage-dependent Ca^2+^ channel, N-Methyl-D-Aspartate (NMDA) receptor and transient receptor potential (TRP). At resting state, the Ca^2+^ level is restored by efflux of Ca^2+^ through plasma membrane channels (Ca^2+^-ATPase, sodium calcium exchanger), binding to Ca^2+^ sensor proteins (calmodulin, calcineurin, calretinin) and uptake by organelles (endoplasmic reticulum and mitochondria) [26]. Calmodulin (CALM) is an important calcium-sensing protein that modulates the presynaptic release of neurotransmitter thru activation of calmodulin-dependent kinase-IIα (CAMKIIα) and CAMKIIβ. In addition, CALM was also reported to control the function of pre-and postsynaptic Ca^2+^ channels [27], synaptic plasticity [28] and modify gene expression [29]. In the present study, TRF treatment of diff-neural cells caused significant upregulation of CALM proteins, namely CALM1, CALM2 and CALM3 proteins (Figure 5, CaM=CALM). The CALM knockdown studies using lentiviral-delivered short-hairpin RNAs have reported no significant changes in neuronal survivability and synapse formation. However, neurons with low levels of the CALM protein, exhibited a spontaneous depression of neuronal network activity [27]. Since CALM proteins are copiously found in the human brain and overexpression studies are relatively scarce, it is difficult to interpret the physiological overexpression of these proteins in our study. Hence, future studies should be directed to understanding the effects of CALM overexpression in modulating the calcium signalling pathway and activity of CAMKII proteins.

Another critical protein that TRF differentially regulated in the calcium signalling pathway is the endoplasmic reticulum chaperone Bip (HSPA5/GRP78/Bip). The HSPA5 protein was significantly down-regulated by TRF compared to untreated controls. No significant changes of this protein were noted in levodopa treated diff-neural cells (Figure 5, Bip=HSPA5). The HSPA5 is a critical protein in the unfolded protein response (UPR) mechanism, an adaptive response that localizes in the endoplasmic reticulum (ER). The UPR is mediated by three ER local proteins: IRE1, PERK and AT6 [30]. At relaxed conditions, HSPA5 binds with these three localized proteins and remain inactive. During ER stress, in which is triggered by disruption in Ca^2+^ homeostasis and redox state and cellular energy supply, HSPA5 dissociates with IRE1, PERK and ATF6. It stimulates the UPR signalling pathway to exert their roles in proteolytic processing of unfolded proteins to prevent aggregation of these proteins [31]. A substantial body of evidence has reported that UPR is activated in post mortem human brain samples of PD patients, indicating ER stress that leads to neurodegeneration [32]. Nevertheless, compelling data from a recent study suggested that HSPA5 mRNA level was significantly upregulated in caudate nucleus, cingulate gyrus, prefrontal and parietal cortex regions of post mortem tissue obtained from PD patients [33]. The suppression of HSPA5 protein has been identified as one of the therapeutic target models in PD. In this regard, Hu and their team have revealed that luteolin, a naturally occurring flavonoid compound, has immense potency in attenuating the 6-OHDA induced augmentation of HSPA5 and down-regulating the ubiquitin proteosome activity. The suppression of HSPA5 protein and UPS signal is one of the neuroprotective approaches in treating neurodegenerative diseases, i.e., PD and AD [34]. Therefore, we suggest that the down-regulation of HSPA5 in diff-neural cells treated with TRF may imply the presence of antioxidant effects that resulted in the absence of ER stress, which kept the HSPA5 at considerably lower concentration in neural cells compared to untreated control cells.

The mitochondria are the principal site for energy metabolism via the Krebs cycle, oxidation phosphorylation and fatty acid β-oxidation pathways. Dysfunction in any of these pathways could compromise the energy synthesis in the form of ATP and cause degeneration of neurons in PD [35]. The exposure of diff-neural cells to TRF yielded a significant upregulation of two significant proteins involved in energy synthesis, which are SLC25A6/ANT3 (ADP/ATP translocase 3) and ATP5F1A (ATP synthase F1 subunit α) (Figure 5, mPTP=SLC25A6, CxV=ATP5F1A). Intriguingly, our data also revealed that diff-neural cells exposed to L-DOPA, displayed a significant upregulation of these two proteins compared to untreated controls. The SLC25 proteins are the largest transporters with 26 members (SLC25A1-46) found abundantly in brain regions such as the hippocampus, thalamus and brainstem, are essential proteins for energy production in neurons [36]. Particularly, the SLC25A6, which is an inherent carrier protein located at the mitochondrial inner membrane and functions as mitochondrial carriers that allow the electrogenic exchange of two native substrates, namely ADP and ATP between mitochondria matrix and intermembrane gap [37]. A study on cancer cells revealed that the SLC25A6 antiport plays a critical role by increasing the metabolic demands for DNA and protein synthesis [38]. Previous studies have delineated that dysfunction in SLC carriers caused fatal neurological disorders. Mutation and decreased SLC30A10 protein levels were discovered in the brain autopsy of Alzheimer’s Disease and PD patients [39]. Besides, an allelic variant in the SLC6A3 gene has been associated with modulation in dopamine metabolism in PD [40]. However, no studies have reported the effect of SLC25A6 protein dysfunction in neuronal cells.

On the other hand, ATP5F1A is found in mitochondria’s cristae and inner membrane and functions in synthesis of ATP from the precursor molecules, ADP and Pi. According to Song et al. [41] the ATP5F1A overexpression may enhance genes related to immune response, angiogenesis and collagen catabolic processes. This study also demonstrated that ATP5F1A might interfere with alternative splicing of genes associated with glucose homeostasis. Notably, a recent study has documented that a lower level of ATP5F1A was detected in circulating small vascular vesicles from PD patients [42]. Along this line, using whole exome sequencing technology, Jonckheere et al. [43] have discovered a heterogenous mutation in ATP5A1 gene in two siblings with severe neonatal encephalopathy. Of note, gene expression profiling of substantia nigra autopsy from PD patients has shown that the ATP5F1A gene was significantly suppressed compared to the control group.

Tubulin is a heterodimer protein that contains a pair of polypeptide chains known as a monomer. It appears as a hollow fibre that polymerizes into long filaments made of α/β tubulin heterodimers that form microtubules [44]. Microtubules are a pertinent component of the cell cytoskeleton that mediate internal trafficking, cell mitosis and repositioning of neurons [45]. A total of 13 protofilaments of microtubules exist but only certain protofilaments can be detected in specialized cells or organisms [46]. We detected six protofilaments of microtubules specific to neural cells, namely TUBA1B, TUBB, TUBB2B, TUBB3, TUBB4B and TUBB6 in our studies. All of these tubulins were significantly upregulated in TRF-treated and levodopa-treated diff-neural cells compared with untreated controls (Figure 3). Microtubule defects have been known to cause neurodegeneration in PD and AD and this impairment is known as axonal transport defects as curated in KEGG-PD pathway [47] (Figure 6). The axonal transport machinery consists of four elements-microtubules, kinesin, dynein and neurofilaments. The microtubules extend along with the axon and transport cargoes of proteins, lipids, RNAs and organelles between soma and distal axonal terminals. Kinesin and dynein are motor proteins that aid in moving the cargoes along the microtubules. Neurofilaments are abundant in axons and play critical roles in preserving axonal stability, transport and shape. Genetic mutation in any component of the axonal transport system is linked to neurodegenerative diseases [48]. Increased expression of TUBB3 in neurons is an indication that differentiating neurons have achieved morphologically and functionally established neuron states [49]. Consistent with this finding, Guo et al. [50] have investigated the roles of TUBB1, TUBB2 and TUBB3 in differentiating neuroblastoma cells. According to this report, TUBB1 and TUBB3 exist in cell bodies and neurites, whereas TUBB2 occurs predominantly in neurites. The gene knock down experiment revealed TUBB1 has a profound effect on cell viability, TUBB2 for neurite outgrowth and TUBB3 for neuroprotection against reactive oxygen species and free radicals. Hence, our study’s increased expression of tubulins in TRF or levodopa treated diff-neural cells may have contributed to enhanced microtubule reorganisation and stabilization that occur in matured differentiated neuronal cells. These functions are important to preserve the physiology and morphology of neurons.

## 5. Conclusions

The label-free mass spectrometry quantification revealed that TRF exhibited a significant role at the molecular level in protecting neuronal cells. We also showed that the TRF regulated key proteins overlaps with levodopa treated neural cells, which indicate the manifestation of common neuroprotective pathways between TRF and levodopa. The KEGG pathway enrichment analysis showed that the Parkinson’s disease (hsa05012) pathway was enriched with the most populated proteins found to be regulated by TRF and levodopa treatment. Upon further investigation into the KEGG-Parkinson’s disease pathway, we discovered that TRF significantly altered the expression of proteins that play a crucial role in the ubiquitin-proteasome pathway, calcium signalling pathway, protein processing in the endoplasmic reticulum, mitochondrial pathway and axonal transport system when compared with levodopa.

## Figures and Tables

**Figure 1 nutrients-14-04632-f001:**
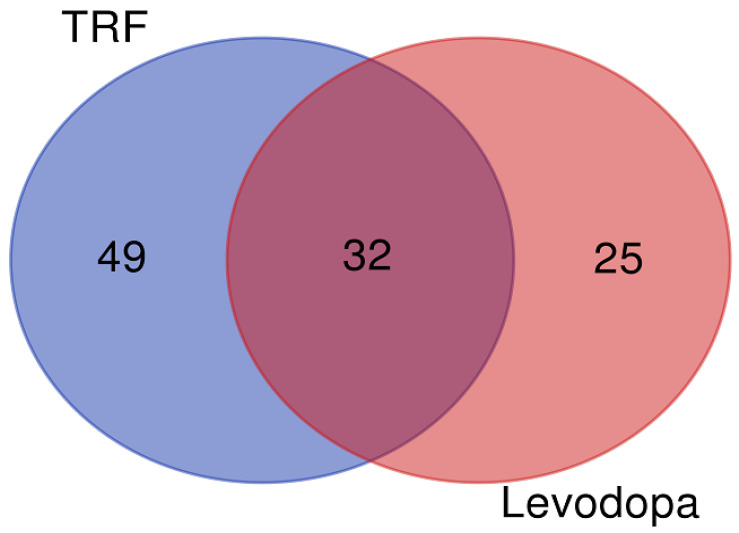
The Venn diagram. The union of intersection of differentially expressed proteins *(p* < 0.05) in diff-neural cell in response to TRF and levodopa treatments.

**Figure 2 nutrients-14-04632-f002:**
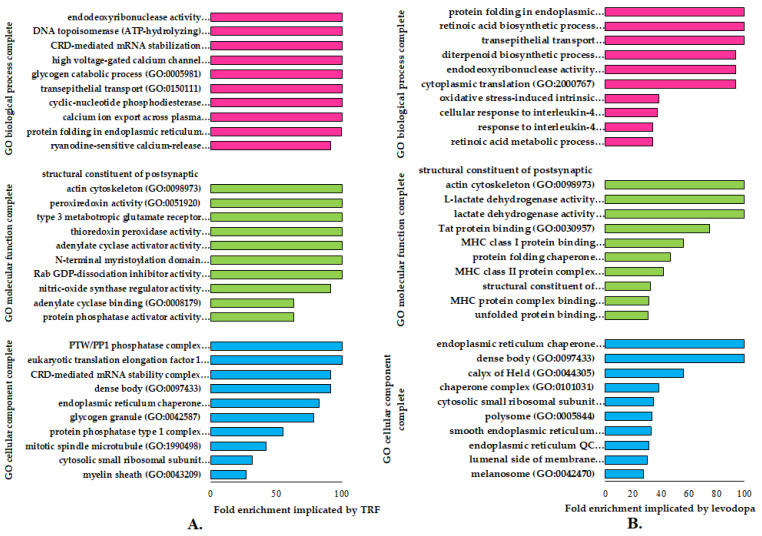
The functional gene ontology (GO) terms of molecular function, biological process and cellular component using PANTHER (version 16) database on differentially regulated proteins (*p* < 0.05) in (**A**) TRF and (**B**) levodopa treatments on diff-neural cells. The horizontal coordinate (x-axis) represents the logarithmic transformation at the base of 10 of the raw *p*-value, and the vertical (Y-axis) displays the GO term classifications.

**Figure 3 nutrients-14-04632-f003:**
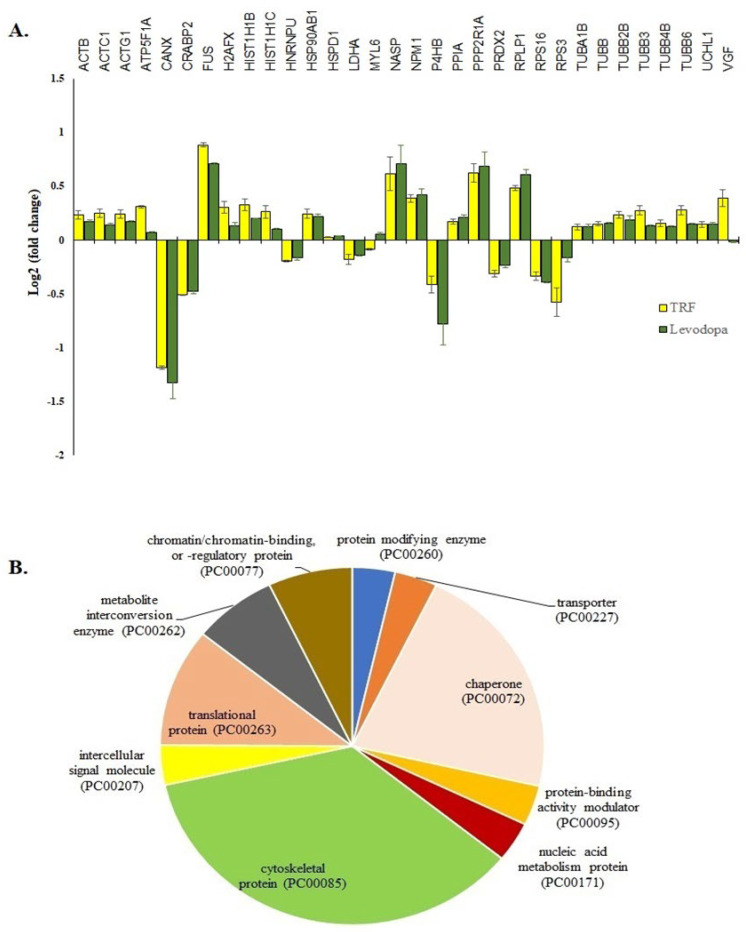
The 32 common proteins expressed in TRF and levodopa treated diff-neural cells. (**A**) Histogram shows the different fold change expressions of the 32 common proteins (*p* < 0.05) regulated by treatment of levodopa or TRF on diff-neural cells. The horizontal coordinate (x-axis) represents the types of protein, and the vertical coordinate (Y-axis) denotes the difference in fold change (logarithmic transformation at the base of 2). (**B**) Pie chart shows the classification of 32 proteins into 10 functional categories using PANTHER online database.

**Figure 4 nutrients-14-04632-f004:**
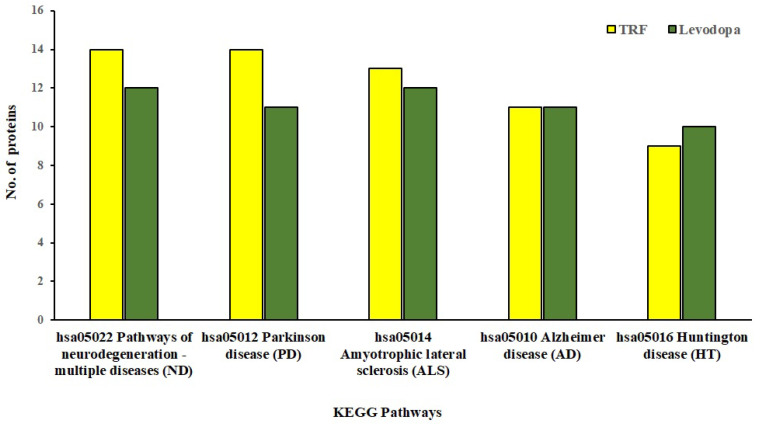
KEGG Pathway enrichment analysis of significantly regulated proteins (*p* < 0.05) by TRF and levodopa on diff-neural cells. Bar chart shows the TRF and levodopa regulated top 5 enriched neurodegenerative pathways.

**Figure 5 nutrients-14-04632-f005:**
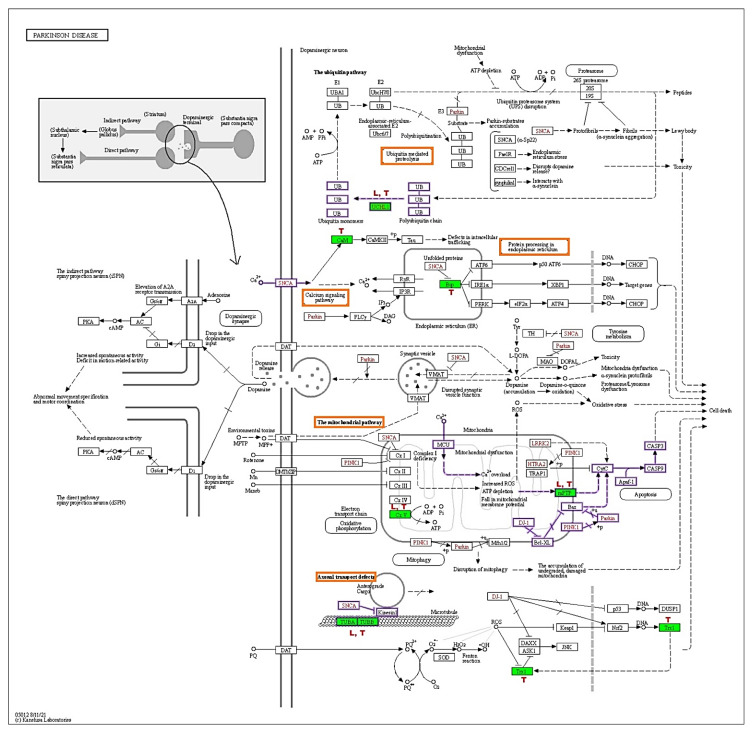
KEGG Parkinson disease pathway (hsa05012). The red boxes indicate the pathways involved by TRF and levodopa, green shaded boxes are proteins that were differentially regulated by (T) TRF and (L) levodopa in differ-neural cells. The proteins behind the KEGG symbol are as follows: UCHL1:UCHL1, CaM: CALM1, CALM2, CALM3, Bip: HSPA5, mPTP: SLC25A6, VDAC1, VDAC3, CxV: ATP5F1A, TUBA/TUBB: TUBA1B, TUBB, TUBB2B, TUBB3, TUBB4B, TUBB6 (Adapted from KEGG-Parkinson’s disease pathway- hsa05012 with permission from Ref. [11]. 2019. Kanehisa, M.).

**Figure 6 nutrients-14-04632-f006:**
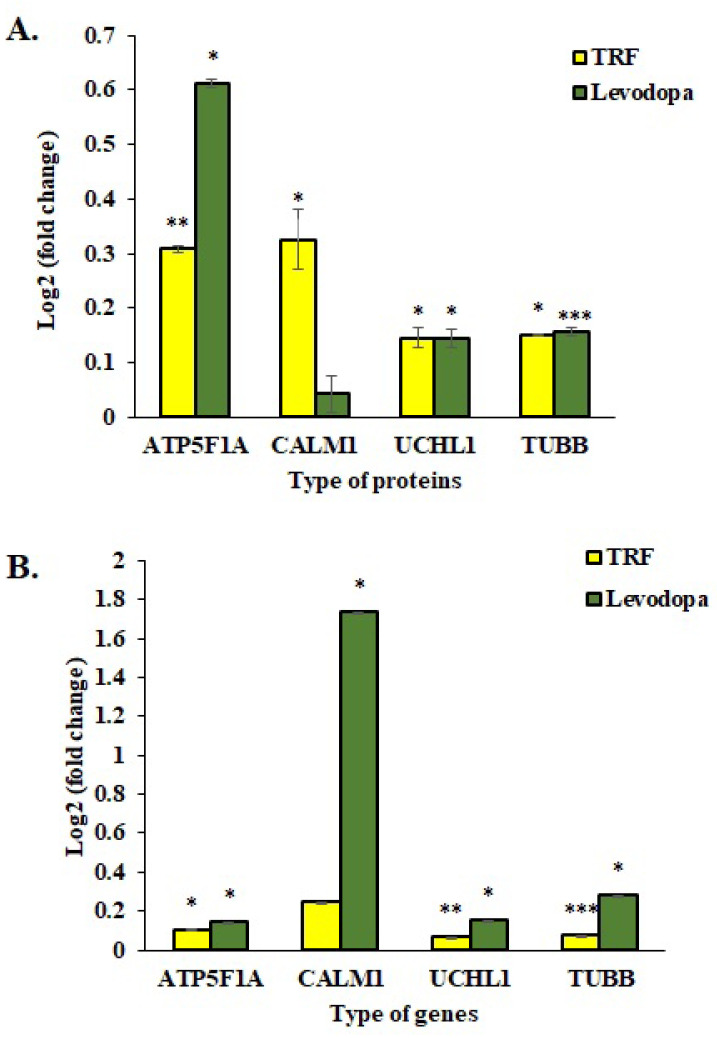
Validation study of mRNA expressions of four selected genes, ATP5F1A, CALM1, UCHL1 and TUBB. (**A**) The log_2_ fold change protein expression following TRF and levodopa treatment on diff-neural cells. (**B**) The log_2_ fold change values of selected mRNA expression following TRF and levodopa treatment on diff-neural cells. For calculation of *p* value, the normalized data set from TRF and levodopa treated group were compared against normalized data obtained from untreated control group. The values are expressed as mean ± SEM (n = 3), Significant difference * *p* < 0.05, ** *p* < 0.01, *** *p* < 0.001 compared to untreated control group.

**Table 1 nutrients-14-04632-t001:** Differentially expressed total protein extract from TRF-treated diff-neural cells compared to untreated control.

Upregulated Proteins					
Uniprot Accession	Description	Symbol	Average Mass	*p* Value	* Log2 (Fold Change)
Q86VP6	Cullin-associated NEDD8-dissociated protein 1	CAND1	136,375	0.021	1.287
Q9UL46	Proteasome activator complex subunit 2	PSME2	27,402	0.005	0.967
P35637	RNA-binding protein FUS	FUS	53,426	0.003	0.885
O43143	Pre-mRNA-splicing factor ATP-dependent RNA helicase	DHX15	90,933	0.010	0.666
P49321	Nuclear autoantigenic sperm protein	NASP	85,238	0.026	0.618
P05386	60S acidic ribosomal protein P1	RPLP1	11,514	0.006	0.484
P14174	Macrophage migration inhibitory factor	MIF	12,476	0.018	0.432
O15240	Neurosecretory protein VGF	VGF	67,258	0.047	0.392
P06748	Nucleophosmin	NPM1	32,575	0.003	0.387
P12236	ADP/ATP translocase 3	SLC25A6	32,866	0.020	0.369
P50395	Rab GDP dissociation inhibitor beta	GDI2	50,663	0.002	0.342
P16401	Histone H1.5	HIST1H1B	22,580	0.030	0.331
P0DP23	Calmodulin-1	CALM1	16,838	0.025	0.326
P0DP24	Calmodulin-2	CALM2	16,838	0.025	0.326
P0DP25	Calmodulin-3	CALM3	16,838	0.025	0.326
Q00610	Clathrin heavy chain 1	CLTC	191,613	0.012	0.319
P52209	6-phosphogluconate dehydrogenase decarboxylating	PGD	53,140	0.032	0.312
P25705	ATP synthase subunit alpha mitochondrial	ATP5F1A	59,751	0.005	0.308
P62841	40S ribosomal protein S15	RPS15	17,040	0.005	0.308
P16104	Histone H2AX	H2AFX	15,145	0.031	0.303
Q71UI9	Histone H2A.V	H2AFV	13,163	0.007	0.296
P0C0S5	Histone H2A.Z	H2AFZ	13,553	0.007	0.296
Q9BUF5	Tubulin beta-6 chain	TUBB6	49,857	0.022	0.277
Q13509	Tubulin beta-3 chain	TUBB3	50,433	0.024	0.276
P16403	Histone H1.2	HIST1H1C	21,365	0.044	0.264
P31150	Rab GDP dissociation inhibitor alpha	GDI1	50,583	0.013	0.250
P68032	Actin alpha cardiac muscle 1	ACTC1	42,019	0.022	0.248
P08238	Heat shock protein HSP 90-beta	HSP90AB1	83,264	0.025	0.244
P63261	Actin cytoplasmic 2	ACTG1	41,793	0.024	0.240
P00558	Phosphoglycerate kinase 1	PGK1	44,615	0.016	0.239
Q9BVA1	Tubulin beta-2B chain	TUBB2B	49,953	0.018	0.236
P60709	Actin cytoplasmic 1	ACTB	41,737	0.025	0.234
P46782	40S ribosomal protein S5	RPS5	22,876	0.022	0.232
P07602	Prosaposin	PSAP	58,113	0.014	0.224
P29401	Transketolase	TKT	67,878	0.043	0.211
P10599	Thioredoxin	TXN	11,737	0.012	0.201
O00571	ATP-dependent RNA helicase DDX3X	DDX3X	73,244	0.007	0.199
P37802	Transgelin-2	TAGLN2	22,391	0.031	0.197
P27348	14-3-3 protein theta	YWHAQ	27,764	0.023	0.185
P62937	Peptidyl-prolyl cis-trans isomerase A	PPIA	18,012	0.024	0.17
P68371	Tubulin beta-4B chain	TUBB4B	49,831	0.035	0.157
P07437	Tubulin beta chain	TUBB	49,671	0.013	0.151
P09936	Ubiquitin carboxyl-terminal hydrolase isozyme L1	UCHL1	24,824	0.043	0.145
P33778	Histone H2B type 1-B	HIST1H2BB	13,950	0.015	0.130
P06899	Histone H2B type 1-J	HIST1H2BJ	13,904	0.015	0.130
P23527	Histone H2B type 1-O	HIST1H2BO	13,906	0.015	0.130
P68363	Tubulin alpha-1B chain	TUBA1B	50,152	0.048	0.124
P63104	14-3-3 protein zeta/delta	YWHAZ	27,745	0.013	0.039
P62136	Serine/threonine-protein phosphatase PP1-alpha catalytic subunit	PPP1CA	37,512	0.040	0.037
P62140	Serine/threonine-protein phosphatase PP1-beta catalytic subunit	PPP1CB	37,187	0.040	0.037
P36873	Serine/threonine-protein phosphatase PP1-gamma catalytic subunit	PPP1CC	36,984	0.040	0.037
P07737	Profilin-1	PFN1	15,054	0.004	0.037
P10809	60 kDa heat shock protein mitochondrial	HSPD1	61,055	0.034	0.025
**Down-Regulated Proteins**					
**Uniprot Accession**	**Description**	**Symbol**	**Average Mass**	***p* value**	*** Log2 (Fold Change)**
P24534	Elongation factor 1-beta	EEF1B2	24,764	0.011	−0.055
P60660	Myosin light polypeptide 6	MYL6	16,930	0.012	−0.088
Q16555	Dihydropyrimidinase-related protein 2	DPYSL2	73,503	0.012	−0.121
Q05639	Elongation factor 1-alpha 2	EEF1A2	50,470	0.035	−0.125
P14625	Endoplasmin	HSP90B1	92,469	0.023	−0.132
P00338	L-lactate dehydrogenase A chain	LDHA	36,689	0.041	−0.178
Q00839	Heterogeneous nuclear ribonucleoprotein U	HNRNPU	90,585	0.017	−0.195
P30041	Peroxiredoxin-6	PRDX6	25,035	0.035	−0.200
P39687	Acidic leucine-rich nuclear phosphoprotein 32 family member A	ANP32A	19,997	0.013	−0.248
P32119	Peroxiredoxin-2	PRDX2	21,892	0.001	−0.312
Q08211	ATP-dependent RNA helicase A	DHX9	140,958	0.041	−0.324
P62249	40S ribosomal protein S16	RPS16	16,445	0.014	−0.336
P19338	Nucleolin	NCL	76,615	0.000	−0.364
Q02878	60S ribosomal protein L6	RPL6	32,728	0.000	−0.366
Q06830	Peroxiredoxin-1	PRDX1	10,676	0.033	−0.394
P11021	Endoplasmic reticulum chaperone BiP	HSPA5	72,333	0.005	−0.405
P09429	High mobility group protein B1	HMGB1	24,894	0.002	−0.414
P07237	Protein disulfide-isomerase	P4HB	57,116	0.033	−0.415
P22626	Heterogeneous nuclear ribonucleoproteins A2/B1	HNRNPA2B1	37,430	0.027	−0.451
P60842	Eukaryotic initiation factor 4A-I	EIF4A1	46,154	0.015	−0.459
P35232	Prohibitin	PHB	22,271	0.002	−0.462
P29373	Cellular retinoic acid-binding protein 2	CRABP2	15,693	0.001	−0.508
P40939	Trifunctional enzyme subunit alpha mitochondrial	HADHA	86,372	0.032	−0.510
P05387	60S acidic ribosomal protein P2	RPLP2	11,665	0.031	−0.521
P23396	40S ribosomal protein S3	RPS3	26,688	0.045	−0.580
P46781	40S ribosomal protein S9	RPS9	22,591	0.036	−0.586
P35579	Myosin-9	MYH9	226,530	0.045	−0.727
P27824	Calnexin	CANX	67,568	0.001	−1.187

* Fold change was estimated by dividing the protein expression in TRF treatment with untreated diff-neural cells.

**Table 2 nutrients-14-04632-t002:** Differentially expressed total protein extract from levodopa-treated diff-neural cells compared to untreated control.

Upregulated Proteins					
Uniprot Accession	Description	Symbol	Average Mass	*p* Value	* Log2 (Fold Change)
P15121	Aldose reductase	AKR1B1	35,853	0.002	0.928
Q09666	Neuroblast differentiation-associated protein	AHNAK	629,114	0.025	0.870
P35637	RNA-binding protein FUS	FUS	53,426	0.002	0.707
P49321	Nuclear autoantigenic sperm protein	NASP	85,238	0.027	0.707
P30153	Serine/threonine-protein phosphatase 2A 65 kDa regulatory subunit A alpha isoform	PPP2R1A	65,309	0.027	0.689
P05386	60S acidic ribosomal protein P1	RPLP1	11,514	0.005	0.611
P06748	Nucleophosmin	NPM1	32,575	0.010	0.422
P04792	Heat shock protein beta-1	HSPB1	22,783	0.007	0.372
P08238	Heat shock protein 90kDa alpha (Cytosolic) class B member 1 isoform CRA_a	HSP90AB1	83,264	0.007	0.22
P62937	Peptidyl-prolyl cis-trans isomerase A	PPIA	18,012	0.015	0.209
P16401	Histone H1.5	HIST1H1B	22,580	0.000	0.203
Q9BVA1	Tubulin beta-2B chain	TUBB2B	49,953	0.038	0.190
P60709	Actin cytoplasmic 1	ACTB	41,737	0.011	0.169
P63261	Actin cytoplasmic 2	ACTG1	41,793	0.010	0.168
P07437	Tubulin beta chain	TUBB	49,671	0.000	0.156
P07195	L-lactate dehydrogenase B chain	LDHB	36,639	0.005	0.150
Q9BUF5	Tubulin beta-6 chain	TUBB6	49,857	0.003	0.148
P09936	Ubiquitin carboxyl-terminal hydrolase isozyme L1	UCHL1	24,824	0.017	0.145
P68032	Actin alpha cardiac muscle 1	ACTC1	42,019	0.023	0.137
Q71U36	Tubulin alpha-1A chain	TUBA1A	50,136	0.038	0.135
P16104	Histone H2AX	H2AFX	15,145	0.038	0.134
Q13509	Tubulin beta-3 chain	TUBB3	50,433	0.004	0.132
P68371	Tubulin beta-4B chain	TUBB4B	49,831	0.005	0.124
P68363	Tubulin alpha-1B chain	TUBA1B	50,152	0.046	0.122
P04406	Glyceraldehyde-3-phosphate dehydrogenase	GAPDH	36,053	0.007	0.122
P11142	Heat shock cognate 71 kDa protein	HSPA8	70,898	0.008	0.122
P09455	Retinol-binding protein 1	RBP1	15,850	0.004	0.115
P16403	Histone H1.2	HIST1H1C	21,365	0.025	0.098
P68104	Elongation factor 1-alpha 1	EEF1A1	50,141	0.006	0.080
P25705	ATP synthase subunit alpha mitochondrial	ATP5F1A	59,751	0.017	0.066
P60660	Myosin light polypeptide 6	MYL6	16,930	0.046	0.055
P10809	60 kDa heat shock protein mitochondrial	HSPD1	61,055	0.013	0.038
**Down-regulated proteins**					
**Uniprot Accession**	**Description**	**Symbol**	**Average Mass**	***p* Value**	*** Log2 (Fold Change)**
P27824	Calnexin	CANX	67,568	0.010	−1.331
Q01105	Protein SET	SET	33,489	0.001	−1.292
P27797	Calreticulin	CALR	48,142	0.026	−1.006
P62851	40S ribosomal protein S25	RPS25	13,742	0.003	−0.925
P07237	Protein disulfide-isomerase	P4HB	57,116	0.042	−0.777
O75531	Barrier-to-autointegration factor	BANF1	10,059	0.000	−0.775
P35232	Prohibitin	PHB	29,804	0.002	−0.745
Q92928	Putative Ras-related protein Rab-1C	RAB1C	22,017	0.032	−0.538
P29373	Cellular retinoic acid-binding protein 2	CRABP2	15,693	0.005	−0.474
Q15084	Protein disulfide-isomerase A6	PDIA6	48,121	0.022	−0.406
P49327	Fatty acid synthase	FASN	273,424	0.007	−0.398
P62249	40S ribosomal protein S16	RPS16	16,445	0.001	−0.390
P62753	40S ribosomal protein S6	RPS6	28,681	0.023	−0.271
P40227	T-complex protein 1 subunit zeta	CCT6A	58,024	0.020	−0.246
P32119	Peroxiredoxin-2	PRDX2	21,892	0.026	−0.230
P14618	Pyruvate kinase PKM	PKM	57,937	0.039	−0.189
P62424	60S ribosomal protein L7a	RPL7A	29,996	0.043	−0.184
P21796	Voltage-dependent anion-selective channel protein 1	VDAC1	30,773	0.042	−0.165
Q9Y277	Voltage-dependent anion-selective channel protein 3	VDAC3	30,659	0.042	−0.165
P23396	40S ribosomal protein S3	RPS3	26,688	0.027	−0.164
Q00839	Heterogeneous nuclear ribonucleoprotein U	HNRNPU	90,585	0.002	−0.162
P13639	Elongation factor 2	EEF2	95,338	0.018	−0.154
P00338	L-lactate dehydrogenase A chain	LDHA	36,689	0.001	−0.141
P23284	Peptidyl-prolyl cis-trans isomerase B	PPIB	23,743	0.038	−0.080
O15240	Neurosecretory protein VGF	VGF	67,258	0.002	−0.014

* Fold change was estimated by dividing the protein expression in levodopa treatment with untreated diff-neural cells.

**Table 3 nutrients-14-04632-t003:** List of unique and common proteins between the levodopa and TRF treatment on diff-neural cell.

Compartment	Total Proteins	Elements
TRF and levodopa	32	HSPD1, HSP90AB1, RPS16, CANX, LDHA, TUBB, NASP, TUBB6, RPLP1, PPIA, ACTB, TUBB4B, MYL6, HIST1H1B, UCHL1, NPM1, FUS, TUBB3, HIST1H1C, TUBA1B, RPS3, CRABP2, VGF, P4HB, H2AFX, PRDX2, ACTC1, ACTG1, TUBB2B, ATP5F1A, HNRNPU, PPP2R1A
Unique to TRF	49	PRDX6, PPP1CB, PGD, SLC25A6, CALM3, PPP1CC, TXN, RPS9, CALM2, HSP90B1, NCL, HSPA5, CAND1, TKT, YWHAQ, HIST1H2BJ, HNRNPA2B1, DHX15, CLTC, MYH9, HADHA, EEF1B2, TAGLN2, PSME2, MIF, PFN1, RPL6, GDI2, HIST1H2BO, DPYSL2, EIF4A1, H2AFZ, RPS5, GDI1, PPP1CA, HMGB1, HIST1H2BB, YWHAZ, PGK1, DHX9, ANP32A, CALM1, RPLP2, DDX3X, H2AFV, EEF1A2, PSAP, PRDX1, RPS15
Unique to levodopa	25	SET, RPS25, RPS6, HSPB1, PPIB, VDAC3, VDAC1, RAB1C, CCT6A, HSPA8, PHB, EEF2, GAPDH, EEF1A1, PDIA6, AHNAK, BANF1, LDHB, RBP1, TUBA1A, PKM, RPL7A, CALR, AKR1B1, FASN

**Table 4 nutrients-14-04632-t004:** List of TRF and levodopa regulated proteins in the top 5 enriched neurodegenerative pathways.

KEGG-Pathways	Pathway Accession	Differentially Regulated Proteins by TRF	Differentially Regulated Proteins by Levodopa
Pathways of neurodegeneration-multiple diseases	hsa05022	ATP5F1A↑, CALM1↑, CALM2↑, CALM3↑, FUS↑, HSPA5↓, SLC25A6↑, TUBA1B↑, TUBB↑, TUBB2B↑, TUBB3↑, TUBB4B↑, TUBB6↑, UCHL1↑	ATP5F1A↑, FUS↑, TUBA1A↑, TUBA1B↑, TUBB↑, TUBB2B↑, TUBB3↑, TUBB4B↑, TUBB6↑, UCHL1↑, VDAC1↓, VDAC3↓
Parkinson’s disease	hsa05012	ATP5F1A↓, CALM1↑, CALM2↑, CALM3↑, HSPA5↓, SLC25A6↑, TUBA1B↑, TUBB↑, TXN↑, TUBB2B↑, TUBB3↑, TUBB4B↑, TUBB6↑, UCHL1↑	ATP5F1A↑, TUBA1A↑, TUBA1B↑, TUBB↑, TUBB2B↑, TUBB3↑, TUBB4B↑, TUBB6↑, UCHL1↑, VDAC1↓, VDAC3↓
Amyotrophic lateral sclerosis	hsa05014	ACTB↑, ACTG1↑, ATP5F1A↓, FUS↑, HNRNPA2B1↓, HSPA5↓, PFN1↑, TUBA1B↑, TUBB↑, TUBB2B↑, TUBB3↑, TUBB4B↑, TUBB6↑	ACTB↑, ACTG1↑, ATP5F1A↑, FUS↑, TUBA1A↑, TUBA1B↑, TUBB↑, TUBB2B↑, TUBB3↑, TUBB4B↑, TUBB6↑, VDAC1↓
Alzheimer’s disease	hsa05010	ATP5F1A↓, CALM1↑, CALM2↑, CALM3↑, SLC25A6↑, TUBA1B↑, TUBB↑, TUBB2B↑, TUBB3↑, TUBB4B↑, TUBB6↑	ATP5F1A↑, GAPDH↑, TUBA1A↑, TUBA1B↑, TUBB↑, TUBB2B↑, TUBB3↑, TUBB4B↑, TUBB6↑, VDAC1↓, VDAC3↓
Huntington disease	hsa05016	ATP5F1A↓, CLTC↑, SLC25A6↑, TUBA1B↑, TUBB↑, TUBB2B↑, TUBB3↑, TUBB4B↑, TUBB6↑	ATP5F1A↑, TUBA1A↑, TUBA1B↑, TUBB↑, VDAC3↓ TUBB2B↑, TUBB3↑, TUBB4B↑, TUBB6↑, VDAC1↓

The sign “↑” indicates up-regulations and “↓” indicates down regulation of protein expression.

## Data Availability

Not applicable.

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
