# Peer review of "Tocotrienol-Rich Fraction and Levodopa Regulate Proteins Involved in Parkinson’s Disease-Associated Pathways in Differentiated Neuroblastoma Cells: Insights from Quantitative Proteomic Analysis"

_nutrients, 2022, doi:10.3390/nu14214632_

Round 1

Reviewer 1 Report

See my comments in the attached PDF file.

Author Response

Please see file attached

Reviewer 2 Report

1) Getting to the conclusion that TRF exerts profound neuroprotective effects by regulating pathways that hamper the development of PD is too broad to make. To make that conclusion, there is a need to add more data with human patient-derived iPSC-derived neurons, patients' tissues, in vivo models, etc. to make that conclusion. 

2) Figures are hard-to-read. Need to improve the resolution of the text.

2) Fig 5 has too much information and is difficult for readers to understand. 

Author Response

Please see file attached
